# Association between Preconception Dietary Fiber Intake and Preterm Birth: The Japan Environment and Children’s Study

**DOI:** 10.3390/nu16050713

**Published:** 2024-02-29

**Authors:** Takahiro Omoto, Hyo Kyozuka, Tsuyoshi Murata, Toma Fukuda, Hirotaka Isogami, Chihiro Okoshi, Shun Yasuda, Akiko Yamaguchi, Akiko Sato, Yuka Ogata, Yuichi Nagasaka, Mitsuaki Hosoya, Seiji Yasumura, Koichi Hashimoto, Hidekazu Nishigori, Keiya Fujimori

**Affiliations:** 1Fukushima Regional Center for the Japan Environmental and Children’s Study, Fukushima 960-1295, Japan; kyozuka@fmu.ac.jp (H.K.); tuyoshim@fmu.ac.jp (T.M.); t323@fmu.ac.jp (T.F.); hisogami@fmu.ac.jp (H.I.); oko13053@fmu.ac.jp (C.O.); room335@fmu.ac.jp (S.Y.); akiko-y@fmu.ac.jp (A.Y.); asato@fmu.ac.jp (A.S.); yuka-o@fmu.ac.jp (Y.O.); nagasaka@fmu.ac.jp (Y.N.); mhosoya@fmu.ac.jp (M.H.); yasumura@fmu.ac.jp (S.Y.); don@fmu.ac.jp (K.H.); nishigo@fmu.ac.jp (H.N.); fujimori@fmu.ac.jp (K.F.); 2Department of Obstetrics and Gynecology, School of Medicine, Fukushima Medical University, Fukushima 960-1295, Japan; 3Department of Pediatrics, School of Medicine, Fukushima Medical University, Fukushima 960-1295, Japan; 4Department of Public Health, School of Medicine, Fukushima Medical University, Fukushima 960-1295, Japan; 5Fukushima Medical Center for Children and Women, Fukushima Medical University, Fukushima 960-1295, Japan

**Keywords:** birth cohort study, dietary fiber, preterm birth, gut microbiota, vaginal microbiota, preconception care

## Abstract

Preterm birth (PTB) is a leading cause of neonatal morbidity and mortality. Therefore, this study aimed to determine whether preconception dietary fiber intake is associated with PTB. This was a prospective cohort Japan Environmental and Children’s Study (JECS). The study population comprised 85,116 singleton live-birth pregnancies from the JECS database delivered between 2011 and 2014. The participants were categorized into five groups based on their preconception dietary fiber intake quintiles (Q1 and Q5 were the lowest and highest groups, respectively). Multiple logistic regression analysis was performed to determine the association between preconception dietary fiber intake and PTB. Multiple logistic regression analysis revealed that the risk for PTB before 34 weeks was lower in the Q3, Q4, and Q5 groups than in the Q1 group (Q3: adjusted odds ratio [aOR] 0.78, 95% confidence interval [CI] 0.62–0.997; Q4: aOR 0.74, 95% CI 0.57–0.95; Q5: aOR 0.68, 95% CI 0.50–0.92). However, there was no significant difference between preconception dietary fiber intake and PTB before 37 weeks. In conclusion, higher preconception dietary fiber intake correlated with a reduced the risk for PTB before 34 weeks. Therefore, new recommendations on dietary fiber intake as part of preconception care should be considered.

## 1. Introduction

Preterm birth (PTB), defined as delivery after 22 weeks but before 37 weeks of gestation, remains a significant cause of neonatal morbidity and mortality and is a major public health challenge worldwide [1,2,3]. The rate of PTB in most developed countries is generally 5–13% [4]. In the Japanese context, the rate of singleton PTB increased from 3.99% in the late 1970s and early 1980s to 4.47% in the period from 2009 to 2014 [5]. Similarly, European countries experienced an increase in PTB rates from 9.8% in 2000 to 10.6% in 2014 [6]. Despite advances in perinatal medicine, the rates of PTB continue to rise, therefore underscoring the critical need for preventive measures. PTB can result from several mechanisms, including infection, inflammation, uteroplacental problems, uterine overdistension, stress, and immunologic processes. While the exact mechanism remains elusive in many cases, it is believed that the interaction of various risk factors triggers the transition from uterine quiescence to delivery. Systemic inflammation associated with multiple risk factors may contribute to the escalating rates of PTB [4,7].

Dietary fiber is defined as carbohydrate polymers containing three or more monomeric units that undergo neither digestion nor absorption within the human small intestine. Dietary fiber is primarily classified into soluble and insoluble forms [8]. Whole grains, vegetables, fruits, legumes, nuts, and other similar foods represent outstanding sources of dietary fiber. Dietary fiber intake is notably effective in promoting overall health, including benefits such as improved digestive health by preventing constipation and supporting regular bowel movements [8]. It also helps with weight management by providing a sense of fullness and reducing calorie intake. A higher dietary fiber intake is associated with a decrease in the incidence of various diseases, such as coronary heart disease, stroke, hypertension, diabetes, obesity, and certain gastrointestinal diseases [9,10,11]. For example, adequate dietary fiber supplementation plays a significant role in the prevention of pancreatic cancer [12]. It also extends its benefits to pregnancy by reducing the risk of gestational diabetes and even reducing the likelihood of developing pancreatic cancer [13,14]. Some of these positive effects may be attributed to changes in gut microbiota. Dietary fiber intake promotes a diverse microbiota, strengthens the gut barrier, modulates the immune system, and contributes to anti-inflammatory effects, influencing overall gut health and potentially reducing the risk of various health issues [15,16].

Recently, the association between maternal microbiota and obstetric outcomes has gained attention. For instance, a well-known finding is the vaginal microbiota. The vagina hosts its own microbiota, which is usually characterized by a microbiome dominated by *Lactobacillus* species. Bacterial vaginosis is characterized by a diverse vaginal microbiota, and its presence heightens the risk of PTB due to associated inflammation [17]. Additionally, pregnancy triggers significant shifts in the gut microbiota, with an increase in lactic acid bacteria and a decrease in butyrate-producing bacteria, which may affect immune responses [17]. Shiozaki et al. reported that the gut microbiota in women with PTB differed from that in women without PTB [18]. This observation implies that optimizing gut microbiota might have a preventive effect on PTB. In other words, regulating the maternal gut microbiota through some kind of treatment could potentially be expected to prevent PTB. However, the efficacy of probiotics intake during pregnancy for PTB is controversial and inconclusive [19,20,21]. Meanwhile, women in early pregnancy who consumed a vegetarian diet, compared to an omnivorous diet, showed an alteration in gut microbiota composition, with features suggesting alterations in fermentation end products from a mixed acid fermentation towards more acetate/butyrate [22]. Furthermore, a preconception dietary pattern characterized by high protein/fruit intake (e.g., fish, meat, chicken, fruit, and some whole grains) was associated with decreased risk for PTB [23]. Based on these findings, we hypothesized that dietary fiber intake before pregnancy until early pregnancy may help in preventing PTB. Therefore, the current study aimed to explore the association between preconception dietary fiber intake and the risk for PTB.

## 2. Materials and Methods

### 2.1. Study Design

We utilized data from the Japan Environment and Children’s Study (JECS), a birth cohort study initiated in January 2011 and funded by the government. With more than 100,000 participants, it follows pregnant women and their offspring, collecting extensive data on lifestyle, genetics, and environmental exposures. The primary objective of the JECS was to investigate the impact of various environmental factors on the health of mothers and children, providing valuable insights for public health policies and interventions [24,25]. The JECS addresses multiple aspects, including prenatal and postnatal exposures, genetic susceptibility, and socio-environmental determinants, to comprehensively examine the complex interactions that shape child health in Japan. To be eligible for participation in the JECS, participants had to meet the following criteria: (1) residence in the study area at the time of recruitment, with plans to continue living in Japan in the near future; (2) pregnant women expected to deliver between August 2011 and mid-2014 (recruited between January 2011 and March 2014); and (3) the ability to comfortably complete the self-administered questionnaires in Japanese.

The JECS protocol was reviewed and approved by the Ministry of the Environment’s Institutional Review Board of Epidemiological Studies and the Ethics Committees of all participating institutions. The study protocol of the JECS was conducted in accordance with the Declaration of Helsinki and other national regulations and guidelines. Written informed consent was obtained from all participants.

### 2.2. Data Collection

We utilized the dataset released in October 2019 (jecs-ta-20190930). This dataset comprised three types of information: (1) a self-reported questionnaire obtained during the first trimester, which collated data on maternal medical characteristics, such as age, height, pre-pregnancy body weight, smoking status, alcohol drinking status, and a food frequency questionnaire (FFQ); (2) a self-reported questionnaire obtained during the second or third trimester providing information on the FFQ and socioeconomic status, including maternal educational status and household income; and (3) information about obstetric outcomes transcribed from the medical records of each co-operating health care provider. Participants with multiple gestations, miscarriage, stillbirth, or insufficient data were excluded.

### 2.3. Determination of Preconception Dietary Fiber Intake, Obstetric Outcomes, and Confounding Factors

To estimate preconception dietary fiber intake, we utilized the FFQ, a semi-quantitative tool that has been validated for use in large-scale Japanese epidemiological studies [26]. Participants were queried about the frequency of their consumption of various food types from 1 year before pregnancy until the first trimester and during the second or third trimester, allowing us to determine their dietary habits prior to conception. Each FFQ item was assigned a standard portion size. Response options for consumption frequency ranged from “almost never” to “≥7 times/day” for foods like vegetables, fruits, and bread and from “almost never” to “≥10 glasses/day” for beverages such as tea and coffee. Multiplying the intake frequencies by the designated portion sizes, we obtained the nutrient content for each food item from the Standard Tables of Food Composition in Japan 2010. Daily dietary fiber intake was estimated by multiplying the consumption frequency by the sum of the dietary fiber content across all food items.

The primary obstetric outcomes were PTB before 37 weeks (including all cases of PTB, with the lowest gestational age being 22 weeks) and PTB before 34 weeks. The following items were used as confounding factors: maternal age, preconception body mass index (BMI), maternal education, household income, preconception total calorie intake, preconception carbohydrate energy ratio, preconception protein energy ratio, preconception fat energy ratio, total calorie intake during pregnancy, carbohydrate energy ratio during pregnancy, fat energy ratio during pregnancy, dietary fiber intake during pregnancy, parity, maternal smoking status, maternal alcohol consumption, pregnancy with assisted reproductive technology (ART), K6 score at the second or third trimester, hypertensive disorders of pregnancy (HDP), and gestational diabetes mellitus (GDM). Maternal age was categorized into four groups: ≤19, 20–29, 30–39, and ≥40 years. Preconception BMI was categorized into three groups: <18.5, 18.5–24.9, and ≥25.0 kg/m^2^. Maternal education was categorized into four groups: junior high school (<10 years), high school (10–12 years), professional school, junior college, or university (13–16 years), and graduate school (≥17 years). Annual household income was categorized into four groups: JPY < 2,000,000, 2,000,000–5,999,999, 6,000,000–9,999,999, and ≥10,000,000. Preconception total calorie, carbohydrate, protein, and fat intakes were estimated using the FFQ similarly as dietary fiber. The carbohydrate, protein, and fat energy ratios were calculated as carbohydrate and protein contain 4 kcal per 1 g, while fat contains 9 kcal per 1 g. These dietary intakes during pregnancy were calculated the same way. Parity was categorized into primipara or multipara. The participants were asked about their smoking status during the first trimester in a self-reported questionnaire and responded “Never”, “Previously did, but quit before realizing current pregnancy”, “Previously did, but quit after realizing current pregnancy”, and “Currently smoking”. Participants who answered “Currently smoking” were categorized as smokers, whereas others were categorized as non-smokers. The participants were asked about their drinking status during the first trimester; the questions included whether they never drank, previously drank but quit, or are currently drinking. Participants who answered “currently drinking” were categorized as alcohol drinkers, whereas others were categorized as non-drinkers. The method of conception was categorized as natural or ART pregnancy, wherein ART was defined as conception after in vitro fertilization and/or intracytoplasmic sperm injection or cryopreserved, frozen, or blastocyst embryo transfer. The K6 score is a mental health screening tool that assesses psychological distress. It consists of six questions related to anxiety and mood disorders, providing a quick and effective measure of an individual’s mental well-being. Respondents rate the frequency of specific symptoms over the past 30 days. Scores range from 0 to 24, with higher scores indicating a higher likelihood of a mental health condition [27]. Participants with a K6 score ≥13 in the second or third trimester were categorized as experiencing psychological distress. HDP was defined as new-onset hypertension (≥140/90 mmHg) after 20 weeks of gestation. HDP is a significant maternal complication with the potential to lead to adverse outcomes for both mother and fetus, ultimately contributing to PTB [4]. GDM was diagnosed using a 75 g oral glucose tolerance test, and diagnosis was made if any of the following criteria were met: cutoff values of ≥92 mg/dL for fasting plasma glucose, ≥180 mg/dL for plasma glucose at 1 h, and ≥153 mg/dL for plasma glucose at 2 h.

### 2.4. Statistical Analysis

The participants were divided into quintiles based on their preconception dietary fiber intake, ranging from Q1 (lowest intake) to Q5 (highest intake). Maternal characteristics and outcomes were summarized for each quintile group. The Kruskal–Wallis (or one-way analysis of variance) and chi-squared tests were used to compare continuous and categorical variables, respectively. Multiple logistic regression models were employed to calculate adjusted odds ratios (aORs) and 95% confidence intervals (CIs) for PTB before 37 and 34 weeks. The models accounted for several factors including maternal age, preconception BMI, maternal education, household income, preconception total calorie intake, preconception carbohydrate energy ratio, preconception protein energy ratio, preconception fat energy ratio, total calorie intake during pregnancy, carbohydrate energy ratio during pregnancy, fat energy ratio during pregnancy, dietary fiber intake during pregnancy, parity, maternal smoking status, maternal alcohol consumption, ART pregnancy, K6 score at the second or third trimester, HDP, and GDM. SPSS version 27 (IBM Corp., Armonk, NY, USA) was used for statistical analyses. Statistical significance was defined with a threshold set at a *p*-value of less than 0.05.

## 3. Results

In the JECS data, a total of 104,062 fetal records from 2011 to 2014 were included. After excluding 1992 infants from multiple gestations, 1189 participants who had miscarriages, 346 participants with stillbirths, 2123 participants with unknown outcomes of live birth, stillbirth, or miscarriage, and 13,296 participants with insufficient data, a final cohort of 85,116 participants who had singleton live births were eligible for this study. These participants were then categorized into dietary fiber quintile groups (Figure 1).

### Maternal Medical Characteristics and Obstetric Outcomes

Table 1 presents a summary of maternal background and obstetric outcomes based on quintiles of preconception dietary fiber intake. The median (inter-quartile range) preconception dietary fiber intake of each group from Q1 to Q5 was 5.5 (4.5–6.2), 8.0 (7.4–8.5), 10.1 (9.6–10.7), 12.8 (12.0–13.7), and 18.4 (16.3–22.1) g/day, respectively. All maternal background variables listed were significantly different among the quintile groups. However, no significant differences were observed in obstetric outcomes.

Participants in the Q1 group tended to have a younger maternal age, a higher proportion of participants with BMI < 18.5 or ≥25.0 kg/m^2^, lower levels of education and income, a higher number of primiparous women, a higher prevalence of smokers, a lower prevalence of drinkers, and a lower proportion of ART pregnancy. In contrast, participants in the Q5 group tended to have an advanced maternal age, a higher proportion of participants with BMI 18.5–24.9 kg/m^2^, higher education and income levels, a higher number of multiparous women, a lower prevalence of smokers, a higher prevalence of drinkers, and a higher proportion of ART pregnancy. The median total calorie intake, mean protein energy ratio, and mean fat energy ratio increased with increasing dietary fiber intake, whereas the mean carbohydrate energy ratio decreased with increasing dietary fiber intake among the quintile groups. The proportion of participants with a K6 score ≥ 13 in the second or third trimester was lower in the Q3 group and higher in the Q1 and Q5 groups.

Table 2 presents data on the relationship between preconception dietary fiber intake and risk for PTB. Multiple logistic regression analysis showed that, using the Q1 group as a reference, the risk (aOR [95% CI]) for PTB before 34 weeks was lower in the Q3, Q4, and Q5 groups (0.78 [0.62–0.997], 0.74 [0.57–0.95], and 0.68 [0.50–0.92], respectively). However, there was no significant difference between preconception dietary fiber intake and PTB before 37 weeks.

## 4. Discussion

### 4.1. Main Findings

To the best of our knowledge, this is the first large prospective birth cohort study investigating the relationship between preconception dietary fiber intake and PTB. We categorized participants with singleton live births into five groups according to their daily fiber intake from 1 year before to the first trimester of pregnancy. We analyzed the association between preconception dietary fiber intake and the occurrence of PTB. Our findings showed a negative correlation between preconception dietary fiber intake and PTB before 34 weeks. Increasing preconception dietary fiber intake correlated with decreasing risk for PTB before 34 weeks, even after accounting for dietary fiber and other major nutrient intakes during pregnancy as confounding factors. This implies the significance of nutritional status during the preconception period.

### 4.2. Interpretation

Our study focused on dietary fiber intake from 1 year before pregnancy until the first trimester of pregnancy. In recent years, there has been a significant increase in interest in preconception care [28,29]. Preconception care is a comprehensive approach to promoting the health of individuals and couples planning to have a child. It involves a range of interventions and guidance, including the management of prospective parents’ overall health, weight control, support for smoking cessation, and nutritional care. There are significant relationships between dietary habits and nutrition status before pregnancy and maternal and offspring outcomes, such as HDP [30,31,32,33], GDM [34,35,36,37], PTB [23,38], and neonatal neurodevelopment [39]. However, although dietary intervention during pregnancy reduced gestational weight gain and cesarean section rates, it had no significant impact on adverse pregnancy complications, including HDP, GDM, and PTB [40]. Therefore, the effectiveness of post-pregnancy dietary interventions is limited, emphasizing the critical role of preconception care in promoting optimal health for both parents and their offspring. Our findings indicate that preconception dietary fiber intake is a beneficial part of preconception care.

We speculate on several mechanisms underlying the efficacy of preconception dietary fiber intake in reducing the risk for PTB. First, dietary fiber intake may induce alterations in the gut microbiota, consequently activating the immune system. The gut microbiota in women with PTB contained lower levels of *Clostridium* and *Bacteroides* compared with women with full-term births [18]. *Clostridium* species in the gut microbiota strongly induce regulatory T cells (Tregs), which are essential for immunosuppression in the intestinal immune system [41]. The finding highlights the role of *Clostridium* species, especially those in clusters IV and XIVa, as potent inducers of Tregs in the colon. The specific factors from *Clostridium* required for Treg induction remain unknown, but the study suggests a diverse set of metabolites collectively produced by various *Clostridium* strains may be essential for optimal Treg induction [41]. The suppression of Tregs leads to the occurrence of PTB because Tregs are associated with immunological tolerance in pregnant women. A balance in immune regulation is essential during pregnancy to avoid immune reactions against the fetus. Studies suggest that an imbalance in Treg function or number could contribute to pregnancy complications, including PTB. In some cases, inadequate Treg activity may lead to an overactive immune response against the developing fetus, triggering inflammation and potentially contributing to PTB [42,43,44,45]. These findings suggest the importance of *Clostridium*-induced Tregs in the gut microbiota regarding the occurrence of PTB. In addition, oral probiotics containing *Clostridium* for women with a high risk of PTB from early pregnancy had a significant effect on PTB prevention before 32 weeks [21]. However, another systematic review and meta-analysis concluded that there was no evidence supporting a significant influence of probiotics or prebiotics on PTB. Examining 27 studies with 2574 publications, the research finds that probiotic intake during pregnancy neither increases nor decreases the risk for PTB before 34 or 37 weeks [20]. The differences in conclusions from these studies may be attributed to differences in variations in bacterial species present in the administered formulations. Moreover, Dominianni et al. reported that dietary fiber intake was associated with a greater abundance of *Clostridium* in the gut microbiota [46]. This study showed a relationship between dietary fiber intake from fruits and vegetables and the composition of the gut microbiome, specifically observing an elevated presence of *Clostridium*. From these results, we hypothesized that active dietary fiber intake before pregnancy may lead to changes in the gut microbiota, resulting in an increase in *Clostridium*, which induces Tregs involved in immune tolerance during pregnancy, potentially leading to a reduction in PTB. Second, another benefit of gut microbiota alteration is attributed to short-chain fatty acids (SCFAs). SCFAs are produced in the human intestine through anaerobic bacterial fermentation of dietary fiber and undigested carbohydrates by the gut microbiota. The major components of SCFAs are formate, acetate, propionate, and butyrate. SCFAs have various physiological effects, influencing host metabolism and health. The primary function of SCFAs is to act as an energy source for colonocytes, enhance the absorption of electrolytes and water, and play a crucial role in maintaining the integrity of the gut epithelial barrier. Additionally, SCFAs have immunomodulatory properties, promoting Tregs and suppressing inflammation [47]. In terms of the relationship between SCFAs and pregnancy, SCFAs reduce the expression of G-protein-coupled receptor 43 (GPR43) and G-protein-coupled receptor 41 (GPR41) involved in inflammatory pathways in uteroplacental tissues and inhibit neutrophil activity. GPR43 and GPR41 expressions were higher in myometrium and fetal membranes after the onset of labor. In addition, a higher expression of GPR43 was observed in fetal membranes of women delivering preterm with evidence of infection [48,49]. Another study showed a negative association between plasma SCFA and PTB [50]. These findings suggest that SCFAs play a protective role in PTB by regulating inflammatory processes through the suppression of GPR43. Third, in addition to changes in the gut microbiota, changes in the vaginal microbiota may also play a role. A major cause of PTB is intrauterine infection resulting from bacterial vaginosis induced by vaginal dysbiosis [4,51,52,53]. Although the precise mechanism linking bacterial vaginosis and PTB remains unclear, it is likely that bacteria ascend into the uterus either before or early in pregnancy, causing infection and triggering intrauterine inflammation, thereby increasing the risk for PTB [4]. Furthermore, Shivakoti et al. reported that women consuming a diet rich in dietary fiber were less likely to be diagnosed with molecular bacterial vaginosis. In this study, multiple logistic regression revealed that diets with higher total dietary fiber intake were inversely associated with the occurrence of molecular bacterial vaginosis (aOR 0.49, 95% CI 0.24–0.99) [54]. This suggests that pregnancy with favorable vaginal microbiota facilitated by the active intake of dietary fiber from the preconception period could reduce the risk for PTB. Our study findings, which showed a significant difference in PTB before 34 weeks, agree with the well-known knowledge that PTB before 34 weeks is more associated with infections than PTB occurring after 34 weeks [48]. Finally, a high-fiber diet may reduce urinary tract infections, which can cause PTB [51], by improving constipation [9]. This is because constipation affects bladder activity through bladder–bowel cross-sensitization and shared nerve pathways [55]. In summary, we hypothesized several factors for the prevention of PTB, including alterations in the gut and vaginal microbiota, an increase in Tregs induced by *Clostridium* in the gut microbiota, and the role of SCFAs associated with an anti-inflammatory effect by suppressing GPR43.

In Japan, the Ministry of Health, Labour and Welfare recommends an ideal daily intake of dietary fiber of over 24 g [56]; however, the median intake among current Japanese women is less than that, as observed in our study. Therefore, the median value of dietary fiber intake (13.7 g/day) for current Japanese adults (18 years and older) and the midpoint (18.9 g/day) of the recommended 24 g/day were used as reference values to calculate the target values. Dietary fiber intake targets were provided for each gender and age group. For instance, the target dietary fiber intake is 16.92 g/day for women aged 18–29 years and 17.60 g/day for women aged 30–49 years. In addition, the target dietary fiber intake during pregnancy is set to the same as for non-pregnant, non-lactating women [56]. These target dietary fiber intakes closely align with the median preconception dietary fiber intake in the Q5 group of the present study, which was 18.4 (inter-quartile range: 16.3–22.1) g/day. Several meta-analyses have investigated the association between dietary fiber intake and the incidence or mortality of major lifestyle diseases, often showing a negative correlation between both variables, although without a clearly defined threshold [57,58]. Although most developed countries recommend a daily dietary fiber intake of 25–32 g for adult women, it is difficult to realistically achieve these ideal amounts through daily food consumption [8]. Based on these findings, we propose increasing the intake of dietary fiber within a reasonable range for women planning to conceive. Increasing dietary fiber consumption could potentially have beneficial effects on overall health and may be a prudent approach to support pregnancy preparation and potentially reduce the risk of adverse outcomes, such as PTB.

### 4.3. Strengths and Limitations

The strength of this study is that it is the first large-scale prospective cohort study in Japan, making it highly representative of the general pregnant population. While a randomized controlled intervention trial is the preferred study design, conducting a long-term controlled trial to examine overall dietary intake as an exposure is impractical. On the other hand, cohort studies provide several benefits, particularly in understanding the relationships between exposures and outcomes over an extended period. Although not derived from a randomized controlled study, this large cohort study offers a rich dataset for analyzing the complex interactions between preconception exposure and obstetric outcomes.

However, this study had some limitations. First, although the questionnaire in this study covered a wide range of perinatal risk factors, there may be other unknown risk factors for PTB. PTB is a complex phenomenon with multifaceted causes, and various factors contribute to the increased risk of PTB, making its pathophysiology intricate. These unknown risk factors could include genetic predispositions, environmental factors, or socio-economic influences not fully captured by current study methodologies. Second, we did not distinguish between spontaneous and indicated PTB due to the lack of detailed information. Generally, the main reason for indicated PTB is HDP [4]. Therefore, we utilized the occurrence of HDP as a confounding factor in the present study. Third, there is a possibility of recall bias influencing the FFQ data because the questionnaire was answered during the first trimester, and the questionnaire queried participants about their dietary habits from 1 year before pregnancy. Recall bias occurs when these recollections are influenced by factors such as memory limitations, individual perceptions, or external influences, resulting in inaccuracies in the reported information. This bias has the potential to compromise the validity of this study. Moreover, it is important to recognize that the results of this study cannot necessarily be generalized to other countries, given the comparatively low PTB rate in Japan [4,5,6]. In addition, the FFQ used in this study was tailored to Japanese women and was based on the unique food culture of Japan. Therefore, caution should be exercised when extrapolating these results to other ethnic groups or regions with potentially different dietary patterns and PTB prevalence.

## 5. Conclusions

Dietary fiber intake from preconception to the first trimester of pregnancy showed a negative association with the occurrence of PTB before 34 weeks. This may be attributed to alterations in the gut and vaginal microbiota and the anti-inflammatory effect of SCFAs, thus suggesting the importance of preconception dietary fiber intake. Further studies are necessary to validate our findings and establish appropriate recommendations for dietary fiber intake as part of preconception care.

## Figures and Tables

**Figure 1 nutrients-16-00713-f001:**
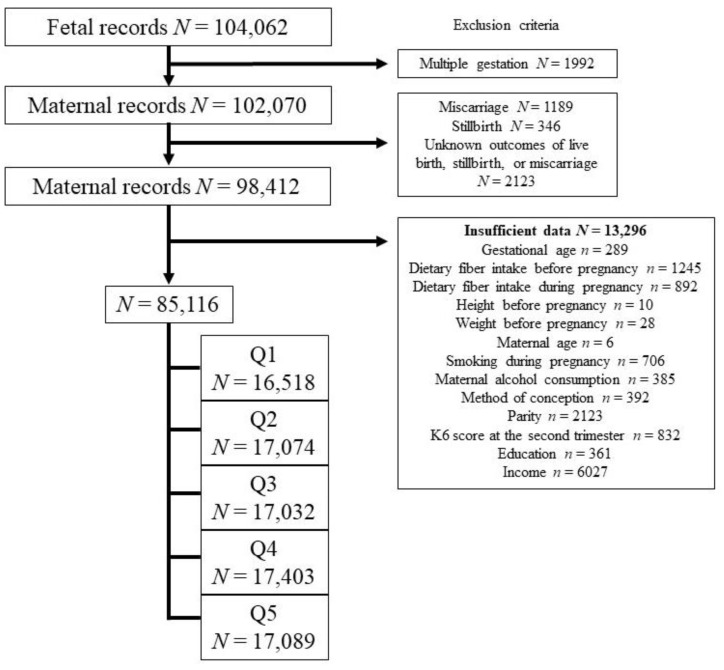
Flowchart of the selection of study participants.

**Table 1 nutrients-16-00713-t001:** Maternal medical background and obstetric outcomes.

	Quintile for Dietary Fiber				
	Q1 (Low)	Q2	Q3	Q4	Q5 (High)
Variable	*n* = 16,518	*n* = 17,074	*n* = 17,032	*n* = 17,403	*n* = 17,089
Maternal medical background					
Preconception dietary fiber intake, g/day median (IQR)	5.5 (4.5–6.2)	8.0 (7.4–8.5)	10.1 (9.6–10.7)	12.8 (12.0–13.7)	18.4 (16.3–22.1)
Maternal age, mean year (SD)	29.8 (5.1)	31.1 (4.9)	31.6 (4.8)	32.0 (4.8)	32.3 (4.7)
Maternal age category, %					
≤19	1.3	0.5	0.3	0.4	0.3
20–29	48.1	38.0	33.6	30.7	28.4
30–39	47.7	57.2	61.5	63.3	65.3
≥40	2.9	4.3	4.5	5.6	5.9
BMI, %					
<18.5	16.5	16.5	16.2	15.4	15.0
18.5–24.9	71.5	72.8	74.0	74.1	74.5
≥25.0	12.0	10.7	9.8	10.5	10.6
Maternal education, years, %					
<10	8.0	4.5	3.7	3.2	3.3
10 to 12	38.7	31.8	29.1	27.2	27.3
13 to 16	52.6	62.4	65.6	67.9	67.3
≥17	0.8	1.3	1.7	1.7	2.0
Household income, JPY, %					
<2,000,000	8.5	5.5	4.6	4.5	5.3
2,000,000–5,999,999	70.5	68.3	67.1	66.1	66.3
6,000,000–9,999,999	18.1	22.5	24.0	24.2	23.2
≥10,000,000	2.9	3.8	4.2	5.2	5.2
Preconception total calorie intake, kcal/day median (IQR)	1198.0 (995.0–1411.0)	1483.0 (1306.0–1695.0)	1686.0 (1479.0–1923.0)	1928.0 (1683.0–2208.0)	2410.0 (2041.0–2942.0)
Preconception carbohydrate energy ratio, mean % (SD)	57.5 (9.9)	55.5 (7.6)	54.7 (7.3)	54.4 (7.0)	53.9 (7.5)
Preconception protein energy ratio, mean % (SD)	12.7 (2.3)	13.3 (1.9)	13.6 (1.9)	13.9 (1.9)	14.3 (2.1)
Preconception fat energy ratio, mean % (SD)	27.3 (8.3)	29.1 (6.4)	29.9 (6.1)	30.3 (5.8)	30.9 (6.1)
Total calorie intake during pregnancy, kcal/day median (IQR)	1274.0 (1036.0–1548.0)	1479.0 (1243.0–1755.0)	1620.0 (1365.0–1920.0)	1783.0 (1502.0–2126.0)	2064.0 (1695.0–2546.0)
Carbohydrate energy ratio during pregnancy, mean % (SD)	57.5 (9.2)	55.8 (7.8)	54.9 (7.5)	54.5 (7.4)	53.8 (7.8)
Protein energy ratio during pregnancy, mean % (SD)	12.8 (2.2)	13.3 (2.0)	13.6 (1.9)	13.9 (1.9)	14.2 (2.1)
Fat energy ratio during pregnancy, mean % (SD)	27.9 (7.9)	29.3 (6.6)	30.0 (6.3)	30.4 (6.2)	31.1 (6.4)
Dietary fiber intake during pregnancy, g/day median (IQR)	6.1 (4.7–7.8)	8.0 (6.6–9.8)	9.6 (7.9–11.6)	11.6 (9.5–14.0)	15.0 (11.9–19.1)
Primipara, %	49.4	42.4	39.4	36.0	32.9
Smoking, %	7.6	4.8	3.9	3.3	3.8
Alcohol, %	8.2	10.1	10.7	10.8	11.0
ART, %	2.1	2.9	3.1	3.4	3.4
K6 score ≥ 13 at the second or third trimester, %	3.6	2.8	2.7	3.0	3.7
Obstetric outcomes					
HDP, %	3.3	3.2	2.9	3.1	3.0
GDM, %	2.9	2.6	2.9	2.7	2.8
PTB < 37 weeks, %	4.7	4.5	4.5	4.3	4.5
PTB < 34 weeks, %	1.0	0.9	0.8	0.8	0.8

ART, assisted reproductive technology; BMI, body mass index; JPY, Japanese yen; HDP, hypertensive disorders of pregnancy; GDM, gestational diabetes mellitus; PTB, preterm birth; SD, standard deviation; IQR, inter-quartile range.

**Table 2 nutrients-16-00713-t002:** Association between dietary fiber intake and the risk of PTB.

	Quintile for Dietary Fiber				
	Q1 (Low)	Q2	Q3	Q4	Q5 (High)
	*n* = 16,518	*n* = 17,074	*n* = 17,032	*n* = 17,403	*n* = 17,089
PTB < 37 weeks					
OR (95% CI)	1 (Ref)	0.96 (0.87–1.06)	0.96 (0.87–1.07)	0.92 (0.83–1.02)	0.96 (0.87–1.06)
aOR (95% CI)	1 (Ref)	0.94 (0.85–1.05)	0.95 (0.85–1.06)	0.90 (0.80–1.01)	0.92 (0.80–1.06)
PTB < 34 weeks					
OR (95% CI)	1 (Ref)	0.85 (0.69–1.06)	0.79 (0.63–0.99) *	0.78 (0.62–0.97) *	0.74 (0.59–0.94) *
aOR (95% CI)	1 (Ref)	0.83 (0.67–1.05)	0.78 (0.62–0.997) *	0.74 (0.57–0.95) *	0.68 (0.50–0.92) *

aOR, adjusted odds ratio; CI, confidence interval; PTB, preterm birth. * *p* < 0.05. Preconception higher dietary fiber intake showed a negative association with the occurrence of PTB before 34 weeks.

## Data Availability

Data are unsuitable for public deposition due to ethical restrictions and the legal framework of Japan. It is prohibited by the Act on the Protection of Personal Information (Act No. 57 of 30 May 2003, amendment on 9 September 2015) to publicly deposit data containing personal information. Ethical Guidelines for Medical and Health Research Involving Human Subjects enforced by the Japan Ministry of Education, Culture, Sports, Science and Technology and the Ministry of Health, Labour and Welfare also restricts the open sharing of the epidemiologic data. All inquiries about access to data should be sent to jecs-en@nies.go.jp. The person responsible for handling enquiries sent to this e-mail address is Shoji F. Nakayama, JECS Programme Office, National Institute for Environmental Studies.

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
