# Peer review of "Association between Preconception Dietary Fiber Intake and Preterm Birth: The Japan Environment and Children’s Study"

_nutrients, 2024, doi:10.3390/nu16050713_

Round 1

Reviewer 1 Report

Comments and Suggestions for Authors

A study of the correlation between preconception dietary fibre intake and the risk of preterm birth is an exciting subject. The authors made a prospective cohort study to demonstrate that the higher preconception dietary fibre intake reduced the risk for PTB before 34 weeks. 

The topic is essential because the results can decrease the number of cases of premature birth, which is a significant public health problem. The study has more limitations than described by the authors. The tables are well-designed and comprehensible.

The conclusions are consistent, and they address the main question. The references are appropriate.

I have some observations:

  1. In Figure 1, 2,123 records were excluded because they were written Unknown. It needs an explanation. 
  2. In the body text, where do you explain that Figure 1 shows that there are 15,419 insufficient data. In the figure, it is written that there are 13,296 insufficient data. It is better to add more explanation to understand the flowchart better. Please make the text “Insufficient data N =13,296” bold. In Figure 1, 289 preterm births are excluded. What is the reason? In these cases, was the gestational age unknown? It needs clarification.
  3. The results are presented related to PTB < 34 weeks and PTB< 37 weeks. We suggest you write the second category as PTB between 34-37 weeks or explain if the group PTB< 37 weeks contains all the premature cases because it is unclear. The article does not describe which limits you are considering for premature birth, and it is essential to present the lower gestational age included in the study.
  4. I recommend moving the Strengths and Limitations section to the end of the Discussion Section.

  I congratulate the authors for the research.

Author Response

1. Summary

Thank you very much for taking the time to review this manuscript. We thank for your thoughtful suggestions and insights. We believe that the comments have helped strengthen the paper considerably. Our point-by-point responses to all the reviewer’s comments are given below. All corresponding changes to the manuscript are indicated in red.

2. Questions for General Evaluation

Reviewer’s Evaluation

Response and Revisions

Does the introduction provide sufficient background and include all relevant references?

Yes

Are all the cited references relevant to the research?

Yes

Is the research design appropriate?

Yes

Are the methods adequately described?

Yes

Are the results clearly presented?

Yes

Are the conclusions supported by the results?

Yes

3. Point-by-point response to Comments and Suggestions for Authors

Comments 1: In Figure 1, 2,123 records were excluded because they were written Unknown. It needs an explanation.

Response 1: Thank you for your suggestion. We have revised the expressions (lines 191-192) and Figure 1.

Comments 2: In the body text, where do you explain that Figure 1 shows that there are 15,419 insufficient data. In the figure, it is written that there are 13,296 insufficient data. It is better to add more explanation to understand the flowchart better. Please make the text “Insufficient data N =13,296” bold. In Figure 1, 289 preterm births are excluded. What is the reason? In these cases, was the gestational age unknown? It needs clarification.

Response 2: Thank you for your suggestion. We have revised the expressions (lines 191-192) and made the “Insufficient data N =13,296” bold in Figure 1. As you indicated, 289 cases were excluded because the gestational age was unknown. I have revised the expression “Preterm birth” to “Gestational age” in Figure 1.

Comments 3: The results are presented related to PTB < 34 weeks and PTB< 37 weeks. We suggest you write the second category as PTB between 34-37 weeks or explain if the group PTB< 37 weeks contains all the premature cases because it is unclear. The article does not describe which limits you are considering for premature birth, and it is essential to present the lower gestational age included in the study.

Response 3: Thank you for your suggestion. We have revised the expressions (lines 129-130).

Comments 4: I recommend moving the Strengths and Limitations section to the end of the Discussion Section.

Response 4: Thank you for your suggestion. We have moved the Strengths and Limitations section to the end of the Discussion Section.

4. Response to Comments on the Quality of English Language

No comment

5. Additional clarifications

None

Reviewer 2 Report

Comments and Suggestions for Authors

Dear authors

what a pleasure, to go trough your manuscript

the topic is of interest and for sure actual

i would like to suggest minor revisions

1) please mention the crucial role that a diet with fiber has to avoid maternal constipation and as a consequence prevent infections (urinary, vaginal) common causes of preterm birth

2) please mention the crucial role that a diet with adequate supplement of fiber has on prevention of pancreatic cancer (read and cite PMID: 34770068) this has a role also during pregnancy in reducing the risk for gestational diabetes and reducting also the risk for pancreatic cancer  read and cite PMID: 34454160

3) please add a summary of your findings in a table

4) please mention about the recommended quantity of fiber during pregnancy as per guidelines suggestion

best regards

Author Response

Response to Reviewer X Comments

1. Summary

Thank you very much for taking the time to review this manuscript. We thank for your thoughtful suggestions and insights. We believe that the comments have helped strengthen the paper considerably. Our point-by-point responses to all the reviewer’s comments are given below. All corresponding changes to the manuscript are indicated in red.

2. Questions for General Evaluation

Reviewer’s Evaluation

Response and Revisions

Does the introduction provide sufficient background and include all relevant references?

Yes

Are all the cited references relevant to the research?

Can be improved

We added several references.

Is the research design appropriate?

Yes

Are the methods adequately described?

Yes

Are the results clearly presented?

Yes

Are the conclusions supported by the results?

Yes

3. Point-by-point response to Comments and Suggestions for Authors

Comments 1: please mention the crucial role that a diet with fiber has to avoid maternal constipation and as a consequence prevent infections (urinary, vaginal) common causes of preterm birth

Response 1: Thank you for your suggestion. We have revised the expressions (lines 321-324) and added Reference 55 (PMID: 23409689).

Comments 2: please mention the crucial role that a diet with adequate supplement of fiber has on prevention of pancreatic cancer (read and cite PMID: 34770068) this has a role also during pregnancy in reducing the risk for gestational diabetes and reducting also the risk for pancreatic cancer  read and cite PMID: 34454160

Response 2: Thank you for your suggestion. We have revised the expressions (lines 55-58) and added References 12 (PMID: 34770068), 13 (PMID: 33693835), and 14 (PMID: 34454160).

Comments 3: please add a summary of your findings in a table

Response 3: Thank you for your suggestion. We have added a summary of our findings in Table 2.

Comments 4: please mention about the recommended quantity of fiber during pregnancy as per guidelines suggestion

Response 4: Thank you for your suggestion. We have revised the expressions (lines 335-336).

4. Response to Comments on the Quality of English Language

No comment

5. Additional clarifications

None
